# POLYAK PARAMETER ENSEMBLE: EXPONENTIAL PARAMETER GROWTH LEADS TO BETTER GENERALIZATION

## ABSTRACT

Building an ensemble model via prediction averaging often improves the generalization performance over a single model across challenging tasks. Yet, prediction averaging comes with three well-known disadvantages: the computational overhead of training multiple models, increased latency and memory requirements at testing. Here, we propose a remedy for these disadvantages. Our approach (PPE) constructs a parameter ensemble model to improve the generalization performance *with virtually no additional computational cost*. During training, PPE maintains a running weighted average of the model parameters at each epoch interval. Therefore, PPE with uniform weights can be seen as applying the Polyak averaging technique at each epoch interval. We show that a weight per epoch can be dynamically determined via the validation loss or pre-determined in an exponentially increasing fashion. We conducted extensive experiments on 11 benchmark datasets ranging from multi-hop reasoning to image classification task. Overall, results suggest that PPE consistently leads to a more stable training and a better generalization across models and datasets.

## 1 INTRODUCTION

Ensemble learning is one of the most effective techniques to improve the generalization of machine learning algorithms (Dietterich, 2000). In its simple form, an ensemble model is constructed from a set of $K$ classifiers over the same training set. At testing time, a new data point is classified by taking a (weighted) average of $K$ predictions (Allen-Zhu & Li, 2023; Sagi & Rokach, 2018). Yet, this setting incurs three disadvantages: the computational overhead of training $K$ models and increased latency and memory requirements at test time (Liu et al., 2022). Here, we address these disadvantages by proposing PPE (Polyak Parameter Ensemble technique) that constructs a parameter ensemble by maintaining a running *weighted* average of the model parameters obtained at each epoch interval. Overall, our extensive experiments on benchmark datasets suggest that PPE consistently leads to a more stable training and a better generalization across models and datasets, e.g., Figure 1 visualizes these improvements on the CIFAR 10 dataset even with uniform ensemble weights.

The idea of averaging parameters of a machine learning model to accelerate stochastic approximation algorithms dates back to Polyak & Juditsky (1992). Averaging parameters obtained in the trajectory of consecutive Stochastic Gradient Descent (SGD) achieves the minimax optimal statistical risk faster than stochastic gradient descent (Jain et al., 2018). Using PPE with positive equal ensemble weights can be seen as using the Polyak averaging technique at each epoch interval. We show that ensemble weights can be either dynamically determined by leveraging the validation loss in a fashion akin to the early stopping (Prechelt, 2002) or pre-determined in an exponentially increasing fashion. Although decreasing the computational cost of using ensemble models at test time has been extensively studied, (Buciluǎ et al., 2006; Garipov et al., 2018; Huang et al., 2017; Wen et al., 2020; Liu et al., 2022), PPE differs from the existing works as PPE does not introduce trainable parameters, does not require an extended training time, and has the same memory the memory requirement of a single model at testing time. Hence, PPE is a cost-free ensemble technique in training and testing time concerned. Overall, our extensive experiments suggest that PPE leads to a more stable training performance (e.g. less zigzaging in the mini-batch losses) and increases the generalizing performances across models, datasets and tasks *at virtually no additional computa-*

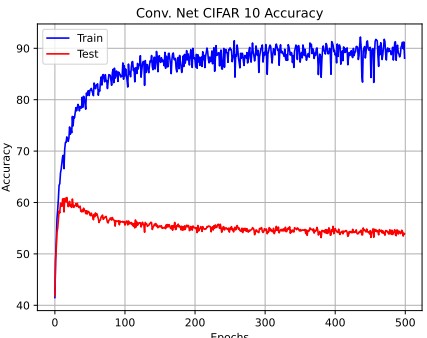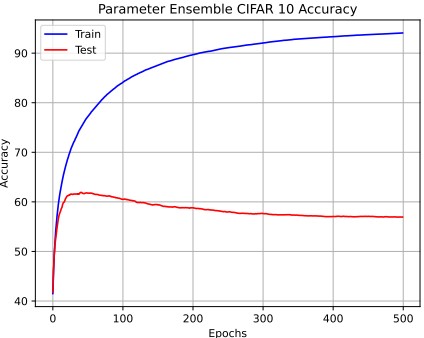

Figure 1: Visualization of train and test accuracy trajectories during the course of the training phase. The figure on the right shows the training and test accuracy of the same network via PPE.

*tional cost.* The benefits of applying PPE becomes more tangible as the number of parameters of a neural model increases, e.g., the embedding size $d$ grows. Although such benefits dissipates at the very low regime ($d \leq 4$) on the benchmark datasets, PPE leads to a better generalization for $d \in \{8, 16, 32, 128, 256\}$.

## 2 RELATED WORK & BACKGROUND

### 2.1 ENSEMBLE LEARNING

Ensemble learning have been extensively studied in the literature (Bishop & Nasrabadi, 2006; Murphy, 2012; Huang et al., 2017; Izmailov et al., 2018). In its simplest form, an ensemble model is constructed from a set of $K$ learners by averaging $K$ predictions (Breiman, 1996). $K$ diverse yet ideally accurate learners are obtained via injecting randomness in the learning process (Allen-Zhu & Li, 2023), e.g., constructing $K$ different training datasets having the same sizes from a given training dataset by randomly sampling data points with replacements or the random initialization of the $K$ models (Goodfellow et al., 2016). At test time, a final prediction is obtained by averaging $K$ predictions. Although this technique introduces the computational overhead of training multiple models and/or increases latency and memory requirements at test time, it often improves the generalization performance in many different learning problems (Murphy, 2012; Goodfellow et al., 2016).

Attempts to alleviate the computational overhead of training multiple models have been extensively studied. For instance, Xie et al. (2013) show that saving parameters of a neural network periodically during training and composing an final prediction via an voting schema stabilizes final predictions. Similarly, Huang et al. (2017) show that using a cosine cyclical learning rate to sample model parameters from different saddle-points improves the performance in multi-class image classification problems. Moreover, the Dropout technique (Srivastava et al., 2014) can be also considered as a form of ensemble learning technique. Dropout prevents the co-adaptation of parameters by stochastically forcing parameters to be zero (Hinton et al., 2012; Srivastava et al., 2014). This technique is regarded as a *geometric averaging* over the ensemble of possible subnetworks (Baldi & Sadowski, 2013). Applying the dropout technique on a single neural network can be interpreted as applying the *Bagging* technique on many neural networks (Goodfellow et al., 2016), i.e., a training procedure for an exponentially large ensemble of networks sharing parameters (Warde-Farley et al., 2013). Monte Carlo Dropout can be seen as a variant of the Dropout that used to approximate model uncertainty in deep learning without sacrificing either computational complexity or test accuracy (Gal & Ghahramani, 2016).

Draxler et al. (2018) and Garipov et al. (2018) independently show that the optima of neural networks are connected by simple pathways having near constant training accuracy. They show that constructing a ensemble model by averaging predictions of such neural networks is a promising means to improve the generalization performance. Although the computational overhead of training multiple models can be alleviated by the aforementioned approaches, the increased latency and memory requirement remain unchanged at test time. For instance, constructing an ensemble of

overparameterized neural models significantly increases the memory requirements (Xu et al., 2021; Demir & Ngonga Ngomo, 2021).

## 2.2 Knowledge Graph Embedding Models and Ensemble Learning

Most Knowledge Graph Embedding (KGE) models are designed to learn continuous vector representations (*embeddings*) of entities and relations tailored towards link prediction/single-hop reasoning (Trouillon et al., 2016; Dettmers et al., 2018; Demir et al., 2021; Ren et al., 2022). They are often formalized as parameterized scoring functions $\phi_{\mathbf{w}} : \mathcal{E} \times \mathcal{R} \times \mathcal{E} \mapsto \mathbb{R}$, where $\mathcal{E}$ denotes a set of entities, and $\mathcal{R}$ stands for a set of relations. $\mathbf{w}$ often consists of a d-dimensional entity embedding matrix $\mathbf{E} \in \mathbb{R}^{|\mathcal{E}| \times d}$ and a d-dimensional relation embedding matrix $\mathbf{R} \in \mathbb{R}^{|\mathcal{R}| \times d}$. Table 1 provides an overview of selected KGE models. Note that we focus on KGE models based on multiplicative interactions, as Ruffinelli et al. (2020) suggest that these multiplicative KGE models often yield state-of-the-art performance if they are optimized well. Moreover, multiplicative KGE models also performs significantly well on multi-hop reasoning tasks (Ren et al., 2023).

Table 1: Overview of KGE models. $\mathbf{e}$ denotes an embedding vector, $d$ is the embedding vector size, $\overline{\mathbf{e}} \in \mathbb{C}$ corresponds to the complex conjugate of $\mathbf{e}_{..}$ $\times_n$ denotes the tensor product along the n-th mode. $\otimes, \circ, \cdot$ stands for Hamilton, Hadamard and inner product, respectively.

| Model | Scoring Function | VectorSpace | Additional |
|---|---|---|---|
| RESCAL Nickel et al. (2011) | $\mathbf{e}_h \cdot \mathcal{W}_r \cdot \mathbf{e}_t$ | $\mathbf{e}_h, \mathbf{e}_t \in \mathbb{R}^d$ | $\mathcal{W}_r \in \mathbb{R}^{d^2}$ |
| DistMult (Yang et al., 2015) | $\mathbf{e}_h \circ \mathbf{e}_r \cdot \mathbf{e}_t$ | $\mathbf{e}_h, \mathbf{e}_r, \mathbf{e}_t \in \mathbb{R}^d$ | - |
| ComplEx (Trouillon et al., 2016) | $\mathrm{Re}(\langle \mathbf{e}_h, \mathbf{e}_r, \overline{\mathbf{e}_t} \rangle)$ | $\mathbf{e}_h, \mathbf{e}_r, \mathbf{e}_t \in \mathbb{C}^d$ | - |
| TuckER Balažević et al. (2019) | $\mathcal{W} \times_1 \mathbf{e}_h \times_2 \mathbf{e}_r \times_3 \mathbf{e}_t$ | $\mathbf{e}_h, \mathbf{e}_r, \mathbf{e}_t \in \mathbb{R}^d$ | $\mathcal{W} \in \mathbb{R}^{d^3}$ |
| QMult (Demir et al., 2021) | $\mathbf{e}_h \otimes \mathbf{e}_r \cdot \mathbf{e}_t$ | $\mathbf{e}_h, \mathbf{e}_r, \mathbf{e}_t \in \mathbb{H}^d$ | - |

Recent results show that constructing an ensemble of KGE models often improves the link prediction performance across datasets and KGE models (Demir et al., 2021; Demir & Ngonga Ngomo, 2021; Xu et al., 2021). Yet, as $|\mathcal{E}|$ grows, leveraging prediction average becomes computational prohibitive as it requires millions of entity embeddings at least twice in memory. KGE models are often trained with one of the three training strategies elucidated below (Dettmers et al., 2018; Lacroix et al., 2018; Ruffinelli et al., 2020).

## 3 Polyak Parameter Ensemble

An ensemble model suffers from expensive memory and computational costs as aforementioned. Here, we introduce PPE, an efficient way to construct a parameter ensemble by maintaining a running *weighted* average of the model parameters obtained at each epoch interval. By leveraging noisy approximations of the gradients during the the mini-batch training regime, we aim to alleviate the necessity of decrease the learning rate to converge a minima as visualized in Figure 2.

By determining the ensemble weights of epochs, we aim to control the *flatness* of this converged minima (Foret et al., 2021). The mini-batch SGD update can be defined as

$$\Theta_{t+1} = \Theta_t - \eta_t \frac{1}{|\mathcal{B}_t|} \sum_b \nabla_\Theta \ell_b(\Theta_t) = \Theta_t - \eta \nabla_\Theta \mathcal{L}_{\mathcal{B}_t}(\Theta_t), \tag{1}$$

where $\mathbf{w}_t \in \mathbb{R}^d, \eta_t > 0$ denotes the learning rate, and $\mathcal{B}_t := \{(\mathbf{x}_b, y_b)\}_{b=1}^m$ is a set of randomly sampled training data points from the training dataset $\mathcal{B} \subset \mathcal{D}$ at a time t. $\nabla_{\mathbf{w}} \ell_b(\mathbf{w}_t)$ denotes the gradient of the loss function on the bases of a single $(\mathbf{x}_b, y_b)$ w.r.t. $\mathbf{w}_t$. Let $\mathbf{w}_t \approx \mathbf{w}^* = \arg\min_{\mathbf{w}} \mathcal{L}_{\mathcal{D}}(\Theta)$ be given. Two consecutive mini-batch SGD updates can be defined as

$$\Theta_{t+1} = \Theta_t - \eta_t \nabla_\Theta \mathcal{L}_{\mathcal{B}_t}(\Theta_t) \tag{2}$$

$$\Theta_{t+2} = \Theta_{t+1} - \eta_{t+1} \nabla_\Theta \mathcal{L}_{\mathcal{B}_{t+1}}(\Theta_{t+1}). \tag{3}$$

A sequence of mini-batch updates leads $\Theta_t$ to hover/circle around $\Theta^*$ provided that $\nabla_\Theta \mathcal{L}_{\mathcal{B}}(\Theta_t) \neq \nabla_\Theta \mathcal{L}_{\mathcal{D}}(\Theta_t)$ and $\eta_t \not\to 0$ as $t \to +\infty$. Since the variance of the fluctuations around $\Theta^*$ is proportional to $\eta$, decreasing $\eta$ is necessary (LeCun et al., 2012).

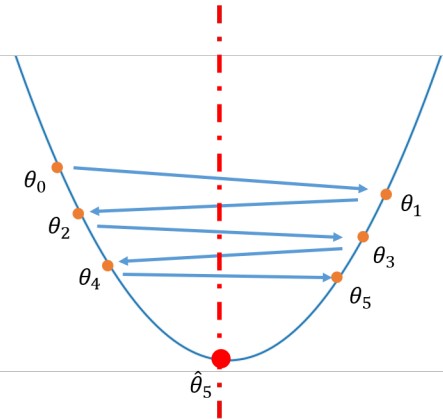

Figure 2: PPE with uniform weights around a minima. $\Theta$ denotes the all trainable parameters.

The former condition of the noisy gradient approximation often holds, since $\mathcal{D}$ does not consist of copies of a single data point (LeCun et al., 2012). Using an optimizer (e.g. Adam) adapting $\eta_t$ w.r.t. $\nabla_\Theta \mathcal{L}_\mathcal{B}(\Theta_t)$ or using a learning rate scheduling technique can often alleviate the issue in practice. Yet, we conjecture that a parameter ensemble model can be constructed from a linear combination of all parameter vectors obtained at each epoch interval. By this, (1) this circling issue can be alleviated regardless of the optimizer and (2) a high performing parameter ensemble is constructed with virtually no additional cost.

Here, we propose PPE (Polyak Parameter Ensemble) technique defined as

$$\Theta_{\text{PPE}} = \text{PPE}(\boldsymbol{\alpha}) = \sum_{i=1}^{N} \boldsymbol{\alpha}_i \odot \Theta_i, \tag{4}$$

where $\Theta_i \in \mathbb{R}^d$ stands for a parameter vector of a model at the end of the i-th epoch, and $\boldsymbol{\alpha}_i$ s.t. $\sum_i^N \boldsymbol{\alpha}_i = 1$ denotes a scalar ensemble weight for the i-th epoch, respectively. $\odot$ multiplies every element of $\Theta_i$ with a scalar ensemble weight $\boldsymbol{\alpha}_i$. Therefore, $\mathbf{w}_{\text{PPE}}$ is a linear combination of all parameter vectors obtained at each epoch. At each epoch $i$, PPE updates the running weighted average of parameters by a scalar matrix multiplication. Setting all $\boldsymbol{\alpha}_{0:N-1} = 0$ and $\boldsymbol{\alpha}_{N-1:N} = 1$ results in obtaining a parameter ensemble model that is only influenced by a parameter vector obtained at the end of the final epoch. Using positive equal ensemble weights $\boldsymbol{\alpha}_i = \frac{1}{N}$ corresponds to applying the Polyak averaging technique at each epoch interval. Next, we show that using such $\boldsymbol{\alpha}$ mitigates the issue of hovering around $\mathbf{w}^*$. A parameter vector $\Theta_{i+1}$ at the end of the i-th epoch can derived as

$$\Theta_{i+1} = \Theta_i - \sum_{t=1}^{T} \eta_{(i,t)} \nabla_\Theta \mathcal{L}_{\mathcal{B}_{(i,t)}}(\Theta_{(i,t)}), \tag{5}$$

where T denotes the number of mini-batches to iterative over the training dataset. $\eta_{(i,t)}$ and $\mathcal{L}_{\mathcal{B}_{(i,t)}}(\Theta_{(i,t)})$ denote the learning rate and the incurred mini-batch loss for the t-th mini-batch step in the i-th epoch, respectively. Assume that $T = 2$ and $N = 2$ with $\eta$, a parameter ensemble model is obtained as follows

$$
\begin{aligned}
\Theta_{\text{PPE}} = \ & \boldsymbol{\alpha}_1\big(\Theta_0 - (\eta_{(0,1)}\nabla_\Theta\mathcal{L}_{\mathcal{B}_{(0,1)}} + \eta\nabla_\Theta\mathcal{L}_{\mathcal{B}_{(0,2)}})\big) + \\
& \boldsymbol{\alpha}_2\big(\Theta_0 - (\eta_{(0,1)}\nabla_\Theta\mathcal{L}_{\mathcal{B}_{(0,1)}} + \eta_{(0,2)}\nabla_\Theta\mathcal{L}_{\mathcal{B}_{(0,2)}} + \\
& \eta_{(1,1)}\nabla_\Theta\mathcal{L}_{\mathcal{B}_{(1,1)}} + \eta_{(1,2)}\nabla_\Theta\mathcal{L}_{\mathcal{B}_{(1,2)}})\big).
\end{aligned}
\tag{6}
$$

where $\nabla_\mathbf{w}\mathcal{L}_{\mathcal{B}_{(i,j)}}$ denotes the gradients of the loss on the bases of the random mini-batch w.r.t. parameter vectors at the i-th epoch and j-th parameter update. Using positive equal ensemble weights $\boldsymbol{\alpha}_1 = \boldsymbol{\alpha}_2 = 1/2$ results in the following parameter ensemble ensemble model

$$
\begin{aligned}
\mathbf{w}_{\text{PPE}} = \Theta_0 - \Big(&\eta_{(0,1)}\nabla_\Theta\mathcal{L}_{\mathcal{B}_{(0,1)}} + \eta_{(0,2)}\nabla_\Theta\mathcal{L}_{\mathcal{B}_{(0,2)}} + \\
& \frac{\eta_{(2,2)}\nabla_\Theta\mathcal{L}_{\mathcal{B}_{(1,1)}}}{2} + \frac{\eta_{(2,2)}\nabla_\Theta\mathcal{L}_{\mathcal{B}_{(2,2)}}}{2}\Big).
\end{aligned}
$$

More generally, using PPE with equal ensemble weights $\boldsymbol{\alpha}_{0:j} = \mathbf{0}$ and $\boldsymbol{\alpha}_{j+1:N} = \frac{1}{\mathbf{N-j}}$ can be rewritten as

$$\Theta_{\text{PPE}} = \Theta_j - \Big( \sum_{i=j+1}^{N} \sum_{t}^{T} \frac{\eta_{(i,t)} \nabla_\Theta \mathcal{L}_{\mathcal{B}_{(i,t)}}}{i} \Big). \tag{7}$$

Therefore, using such ensemble weights (i.e. averaging parameters at each epoch interval) results in deriving such $\mathbf{w}_{\text{PPE}}$ that is more heavily influenced by the parameter vectors obtained at the early stage of the training phase starting from $\mathbf{w}_j$. Consequently, selecting such $j$-th epoch, after $\mathcal{L}_{\mathcal{D}}(\Theta)$ stagnates, the circling issue around a minima can be alleviated. Yet, using positive equal $\boldsymbol{\alpha}$ arguably may not be optimal for all learning problems, since it is implicitly assumed that $\Theta_j \approx \mathbf{w}^*$. In B, we propose two techniques to determine $\boldsymbol{\alpha}$.

### 3.1 DETERMINING ENSEMBLE WEIGHTS $\boldsymbol{\alpha}$

Parameter ensemble weights $\boldsymbol{\alpha}$ can be determined in various fashion. For instance, $\boldsymbol{\alpha}$ can be dynamically determined in a fashion akin to the early stooping technique, i.e., tracking the trajectory of the validation losses at each epoch (Prechelt, 2002). More specifically, initially $\boldsymbol{\alpha}$ are initialized with zeros. As the discrepancy between the validation and training loss exceeds a certain threshold at the end of the $j$-th epoch, positive equal ensemble weights can be used, i.e., $\boldsymbol{\alpha}_{0:j} = \mathbf{0}$ and $\boldsymbol{\alpha}_{j+1:N} = \frac{1}{\mathbf{N-j}}$. Although this may alleviate possible overfitting, hence improve the generalization performance, assigning equal weights for $N - j$ epochs may not be ideal for all learning problems. Instead, remaining parameter ensemble weights $\boldsymbol{\alpha}$ can be determined in an exponentially increasing manner, i.e., $\boldsymbol{\alpha}_{j+1} = \lambda \boldsymbol{\alpha}_j$, where $\lambda$ is a scalar value denoting the rate of increase. More details about the ensemble weights with different growth rates can be found in the supplementary material.

## 4 EXPERIMENTS

### 4.1 TRAINING AND OPTIMIZATION

We followed the experimental setup used by Ruffinelli et al. (2020) for the link prediction task. We trained DistMult, ComplEx, and QMult KGE models with the following hyperparameter configuration: the number of epochs $N \in \{200, 250\}$, Adam optimizer with $\eta = 0.1$, batch size 1024, layer normalization on embedding vectors and an embedding vector size $d \in \{256, 128, 64\}$, and $\lambda \in \{1.0, 1.1\}$. Note that $d = 256$ corresponds to 256 real-valued embedding vector size, hence 128 and 64 complex- and quaternion-valued embedding vector sizes respectively. We ensure that all models have the same number of parameters, while exploring various $d$. Throughout our experiments, we used KvsAll training strategy. For multi-hop query answering, we followed the experimental setup used by Arakelyan et al. (2021). More specifically, we compute query scores for entities via the beam search combinatorial optimization procedure, we keep the top 10 most promising variable-to-entity substitutions.

On all datasets, we used the same the j-th epoch to determine ensemble weights $\boldsymbol{\alpha}$ (see 7) We fixed the $j$-th epoch as 200. Hence, in our experiments, we did not dynamically determined $\boldsymbol{\alpha}$ by tracking the validation loss. By doing this, we ensure that PPE **does not benefit from any additional information (e.g. the validation loss) during the training process**, since tracking the validation loss during training may implicitly leverages information about the generalization performance of a parameter ensemble model.

### 4.2 DATASETS

We used the standard benchmark datasets (UMLS, KINSHIP, NELL-995 h25, NELL-995 h50, NELL-995 h100, FB15K-237, YAGO3-10) for the link prediction and multi-hop query answering tasks. Overviews of the datasets and queries are provided in Table 2 and Table 3, respectively. For the image classification task, we used the CIFAR-10 dataset (Krizhevsky et al., 2009). Further details about the benchmark datasets are relegated to the appendix.

Table 2: An overview of datasets in terms of number of entities, number of relations, and node degrees in the train split along with the number of triples in each split of the dataset.

| Dataset | $|\mathcal{E}|$ | $|\mathcal{R}|$ | $|\mathcal{G}^{\text{Train}}|$ | $|\mathcal{G}^{\text{Validation}}|$ | $|\mathcal{G}^{\text{Test}}|$ |
|---|---|---|---|---|---|
| Mutagenesis | 14,250 | 8 | 55,023 | - | - |
| Carcinogenesis | 22,540 | 9 | 63,382 | - | - |
| UMLS | 135 | 46 | 5,216 | 652 | 661 |
| KINSHIP | 104 | 25 | 8,544 | 1,068 | 1,074 |
| NELL-995 h100 | 22,411 | 43 | 50,314 | 3,763 | 3,746 |
| NELL-995 h50 | 34,667 | 86 | 72,767 | 5,440 | 5,393 |
| NELL-995 h25 | 70,145 | 172 | 122,618 | 9,194 | 9,187 |
| FB15K-237 | 14,541 | 237 | 272,115 | 17,535 | 20,466 |
| YAGO3-10 | 123,182 | 37 | 1,079,040 | 5,000 | 5,000 |

Table 3: Overview of different query types. Query types are taken from Ren et al. (2020).

| Multihop Queries | |
|---|---|
| 2p | $E_? \, . \, \exists E_1 : r_1(e, E_1) \wedge r_2(E_1, E_?)$ |
| 3p | $E_? \, . \, \exists E_1 E_2 . r_1(e, E_1) \wedge r_2(E_1, E_2) \wedge r_3(E_2, E_?)$ |
| 2i | $E_? \, . \, r_1(e_1, E_?) \wedge r_2(e_2, E_?)$ |
| 3i | $E_? \, . \, r_1(e_1, E_?) \wedge r_2(e_2, E_?) \wedge r_3(e_3, E_?)$ |
| ip | $E_? \, . \, \exists E_1 . r_1(e_1, E_1) \wedge r_2(e_2, E_1) \wedge r_3(E_1, E_?)$ |
| pi | $E_? \, . \, \exists E_1 . r_1(e_1, E_1) \wedge r_2(E_1, E_?) \wedge r_3(e_2, E_?)$ |
| 2u | $E_? \, . \, r_1(e_1, E_?) \vee r_2(e_2, E_?)$ |
| up | $E_? \, . \, \exists E_1 . [r_1(e_1, E_1) \vee r_2(e_2, E_1)] \wedge r_3(E_1, E_?)$ |

## 4.3 EVALUATION

We evaluated the link prediction performance of DistMult, ComplEx, and QMult with and without PPE. To this end, we used the Hits@N and MRR benchmark metrics (Ruffinelli et al., 2020). We reported the Hits@N and MRR training, validation and test scores on each benchmark dataset for the link prediction problem. By reporting the training and validation scores, we aim to detect possible impacts on the training performance. Since the Mutagenesis and Carcinogenesis datasets do not contain the validation and test splits, we applied 10-fold cross validated results and reported the mean results.

## 5 RESULTS

Tables 4 to 6 report link prediction results on the link prediction benchmark datasets. We relegate the experiments on UMLS and KINSHIP into the appendix. Overall, our experiments suggest that using PPE consistently improves the link prediction performance of DistMult, ComplEx, and QMult on all datasets. The results of our parameter analysis suggest that as the embedding size $d$ increases, the benefits of using PPE becomes more tangible. Throughout our experiments, we did not detect any runtime overhead of using PPE.

Table 4 reports link prediction results on the FB15K-237 and YAGO3-10 benchmark datasets. Overall, results suggest that PPE improves the generalization performance of DistMult, ComplEx, and QMult on both datasets. Our results also indicate that using PPE improves the link prediction performance even on the training datasets. Despite all models being trained with the Adam optimizer, PPE seems alleviate the circling behavior around a minimum further. Table 5 suggests that PPE improves the generalization performance across models on NELL-995 h25 and NELL-995 h50.

Table 6 reports 10-fold cross validated link prediction results on bio-related benchmark datasets. Since Mutagenesis and Carcinogenesis datasets do not contain a validation and test datasets for the link prediction problem, we conducted experiments with 10-fold cross validation.

Table 4: Link prediction results on the train, validation and test splits of FB15K-237 and YAGO3-10. Bold results indicate the best results.

| | FB15K-237 | | | | YAGO3-10 | | | |
|---|---|---|---|---|---|---|---|---|
| | MRR | @1 | @3 | @10 | MRR | @1 | @3 | @10 |
| DistMult-train | 0.991 | 0.985 | **0.999** | **1.000** | 0.980 | 0.962 | **0.998** | **1.000** |
| With PPE | **0.994** | **0.990** | 0.997 | **1.000** | **0.981** | **0.963** | **0.998** | 0.999 |
| DistMult-val | 0.124 | 0.074 | 0.132 | 0.222 | 0.400 | 0.337 | 0.433 | 0.520 |
| With PPE | **0.138** | **0.082** | **0.149** | **0.249** | **0.446** | **0.384** | **0.481** | **0.558** |
| DistMult-test | 0.122 | 0.071 | 0.129 | 0.223 | 0.393 | 0.330 | 0.425 | 0.512 |
| With PPE | **0.134** | **0.080** | **0.145** | **0.243** | **0.441** | **0.377** | **0.481** | **0.558** |
| ComplEx-train | 0.995 | 0.991 | 0.999 | **1.000** | **0.984** | **0.969** | **1.000** | **1.000** |
| With PPE | **0.996** | **0.993** | **1.000** | **1.000** | **0.984** | **0.969** | **1.000** | **1.000** |
| ComplEx-val | 0.128 | 0.075 | 0.138 | 0.233 | 0.408 | 0.344 | 0.439 | 0.530 |
| With PPE | **0.153** | **0.095** | **0.169** | **0.270** | **0.444** | **0.378** | **0.484** | **0.562** |
| Complex-test | 0.126 | 0.075 | 0.134 | 0.229 | 0.394 | 0.325 | 0.431 | 0.525 |
| With PPE | **0.150** | **0.094** | **0.165** | **0.264** | **0.433** | **0.366** | **0.473** | **0.554** |
| QMult-train | 0.989 | 0.981 | 0.997 | 0.999 | 0.821 | 0.790 | 0.841 | 0.877 |
| With PPE | **0.995** | **0.990** | **1.000** | **1.000** | **0.828** | **0.792** | **0.852** | **0.881** |
| QMult-val | 0.141 | 0.083 | 0.151 | 0.258 | 0.306 | 0.241 | 0.338 | **0.427** |
| With PPE | **0.172** | **0.108** | **0.188** | **0.302** | **0.341** | **0.283** | **0.367** | 0.346 |
| QMult-test | 0.138 | 0.082 | 0.146 | 0.253 | 0.300 | 0.232 | 0.334 | 0.424 |
| With PPE | **0.167** | **0.102** | **0.183** | **0.298** | **0.339** | **0.282** | **0.365** | **0.444** |

Table 5: Link prediction results on the train, validation and test splits of NELL-995 h25 and NELL-995 h50 benchmark datasets. Bold results indicate the best results.

| | h25 | | | | h50 | | | |
|---|---|---|---|---|---|---|---|---|
| | MRR | @1 | @3 | @10 | MRR | @1 | @3 | @10 |
| DistMult-train | **0.995** | 0.991 | 0.998 | **1.000** | **0.955** | **0.934** | **0.974** | **0.990** |
| With PPE | **0.995** | **0.992** | **0.999** | **1.000** | 0.895 | 0.863 | 0.921 | 0.951 |
| DistMult-val | 0.151 | 0.107 | 0.164 | 0.235 | 0.162 | 0.114 | 0.178 | **0.258** |
| With PPE | **0.159** | **0.116** | **0.172** | **0.240** | **0.164** | **0.116** | **0.184** | 0.257 |
| DistMult-test | 0.154 | 0.111 | 0.168 | 0.238 | 0.164 | 0.116 | 0.116 | 0.257 |
| With PPE | **0.162** | **0.119** | **0.177** | **0.245** | **0.166** | **0.119** | **0.184** | **0.258** |
| ComplEx-train | **1.000** | **1.000** | **1.000** | **1.000** | 0.991 | 0.986 | 0.995 | 0.996 |
| With PPE | **1.000** | **1.000** | **1.000** | **1.000** | **0.995** | **0.991** | **0.999** | **1.000** |
| ComplEx-val | **0.105** | **0.069** | **0.110** | 0.175 | 0.079 | 0.048 | 0.082 | 0.143 |
| With PPE | **0.105** | **0.069** | **0.110** | **0.176** | **0.089** | **0.054** | **0.094** | **0.160** |
| Complex-test | **0.106** | **0.071** | **0.110** | 0.175 | 0.080 | 0.049 | 0.085 | 0.141 |
| With PPE | 0.105 | 0.069 | **0.110** | **0.178** | **0.093** | **0.058** | **0.097** | **0.160** |
| QMult-train | 0.977 | 0.967 | 0.986 | 0.994 | 0.917 | 0.890 | 0.934 | 0.962 |
| With PPE | **1.000** | **0.999** | **1.000** | **1.000** | **0.924** | **0.902** | **0.937** | **0.966** |
| QMult-val | 0.084 | 0.055 | 0.090 | 0.140 | 0.102 | 0.058 | 0.115 | 0.191 |
| With PPE | **0.090** | **0.059** | **0.096** | **0.150** | **0.114** | **0.068** | **0.127** | **0.206** |
| QMult-test | 0.081 | 0.052 | 0.086 | 0.138 | 0.105 | 0.061 | 0.114 | 0.191 |
| With PPE | **0.086** | **0.055** | **0.090** | **0.146** | **0.116** | **0.070** | **0.126** | **0.211** |

Table 7 reports the multi-hop queries answering results on the UMLS dataset. Due to the space constraint, we relegate the results of ComplEx and QMult into appendix. Results suggest that PPE on average increases the multi-hop query answering performance across query types.

Table 6: 10-fold cross validated link prediction results on Mutagenesis and Carcinogenesis datasets. Bold results indicate the best results.

| | Mutagenesis | | | | Carcinogenesis | | | |
|---|---|---|---|---|---|---|---|---|
| | MRR | @1 | @3 | @10 | MRR | @1 | @3 | @10 |
| DistMult | 0.150 | 0.121 | 0.151 | 0.204 | 0.045 | 0.025 | 0.047 | 0.085 |
| With PPE | **0.186** | **0.156** | **0.192** | **0.225** | **0.059** | **0.034** | **0.063** | **0.106** |
| ComplEx | 0.143 | 0.109 | 0.148 | 0.200 | 0.054 | **0.027** | 0.056 | 0.110 |
| With PPE | **0.203** | **0.158** | **0.220** | **0.286** | **0.056** | **0.027** | **0.059** | **0.117** |
| QMult | 0.136 | 0.104 | 0.137 | 0.190 | **0.033** | 0.014 | **0.029** | **0.065** |
| With PPE | **0.195** | **0.154** | **0.203** | **0.266** | **0.033** | **0.015** | **0.029** | **0.065** |

Table 7: Query answering results for 8 types of multi-hop queries. Average denotes the average MRR, Hit@1 or Hit@3 scores across all types of queries.

| Method | Average | 2p | 3p | 2i | 3i | ip | pi | 2u | up |
|---|---|---|---|---|---|---|---|---|---|
| | **MRR** | | | | | | | | |
| DistMult | 0.378 | 0.238 | 0.194 | 0.642 | 0.659 | 0.332 | 0.467 | 0.349 | 0.140 |
| With PPE | **0.386** | **0.296** | **0.202** | **0.628** | **0.632** | **0.333** | **0.498** | **0.340** | **0.161** |
| | **HITS@1** | | | | | | | | |
| DistMult | 0.259 | 0.106 | 0.101 | **0.539** | **0.549** | 0.200 | 0.321 | **0.236** | 0.020 |
| With PPE | **0.274** | **0.191** | **0.117** | 0.507 | 0.540 | **0.210** | **0.355** | 0.223 | **0.047** |
| | **HITS@3** | | | | | | | | |
| DistMult | 0.415 | 0.278 | **0.214** | 0.686 | **0.726** | **0.380** | 0.512 | **0.375** | 0.148 |
| With PPE | **0.423** | **0.329** | 0.208 | **0.692** | 0.684 | 0.366 | **0.574** | 0.345 | **0.188** |

Figure 1 visualizes the stable training and testing accuracy achieved via PPE in image classification on the Cifar 10 dataset. Table 8 reports the link prediction results with different embedding dimension sizes. Overall, results suggest that as d grows, the benefits of PPE becomes more tangible. For instance, if $d \geq 32$ DistMult with PPE reaches **81 higher Hit@N or MRR out of 96 scores**, while DistMult with PPE performed slightly worse at only 6 out of 96 scores. Using the 1.1 growth rate leads to a slight improvement over no growth rate.

## 5.1 DISCUSSION

Our results indicate that constructing a parameter ensemble model by maintaining a weighted average of parameters obtained at each epoch interval improves the generalization performance across datasets and knowledge graph embedding models. We show that with our formulation, weights/parameter ensemble weights can be determined in various forms, e.g., dynamically by tracking the validation loss or choosing an exponential function over weights. Overall, results suggest that using exponentially increasing ensemble weights consistently improves the generalization results. This may suggest that although Adam dynamically adapts the learning rate w.r.t. the gradients, our weighted parameter averaging approach (PPE) accelerates the converge on a minima during training. Yet, our parameter analysis show that the benefits of applying PPE dissipates if the embedding vector is very low (e.g. $d < 16$).

## 6 CONCLUSION

In this work, we investigated means to construct a parameter ensemble for knowledge graph embedding models. By this, we aimed to alleviate the three computational disadvantages of constructing an ensemble model based on the prediction averaging technique: the computational overhead of training multiple models and increased latency and memory requirements at test time. We showed that building a high performing parameter ensemble by maintaining a running weighted average

Table 8: Link prediction results of DistMult with different embedding dimensions $d$ on the train, validation and test splits of UMLS and KINSHIP benchmark datasets. PPE[†] denotes applying PPE with $\lambda = 1.1$. Bold results indicate the best results.

| | d | UMLS | | | | KINSHIP | | | |
|---|---|---|---|---|---|---|---|---|---|
| | | MRR | @1 | @3 | @10 | MRR | @1 | @3 | @10 |
| DistMult-test | 2 | 0.309 | 0.225 | **0.321** | **0.470** | **0.061** | **0.010** | **0.039** | **0.123** |
| With PPE | | **0.310** | **0.226** | 0.318 | 0.469 | 0.054 | **0.010** | 0.033 | 0.106 |
| With PPE[†] | | 0.309 | 0.225 | 0.316 | 0.469 | 0.055 | **0.010** | 0.031 | 0.107 |
| DistMult-test | 4 | 0.617 | 0.486 | 0.711 | 0.834 | **0.105** | **0.038** | **0.083** | 0.218 |
| With PPE | | **0.627** | **0.490** | **0.724** | 0.837 | 0.099 | 0.028 | **0.083** | 0.220 |
| With PPE[†] | | 0.626 | 0.489 | 0.723 | **0.840** | 0.102 | 0.033 | 0.082 | **0.221** |
| DistMult-test | 8 | **0.775** | **0.663** | **0.869** | **0.939** | 0.518 | 0.351 | 0.602 | 0.886 |
| With PPE | | 0.763 | 0.643 | 0.865 | 0.935 | 0.533 | 0.373 | **0.616** | 0.883 |
| With PPE[†] | | 0.764 | 0.644 | 0.865 | 0.935 | **0.534** | **0.376** | 0.614 | **0.883** |
| DistMult-test | 16 | 0.862 | 0.781 | **0.936** | 0.981 | 0.665 | 0.520 | 0.771 | **0.938** |
| With PPE | | 0.863 | 0.785 | 0.932 | 0.982 | 0.669 | 0.523 | 0.779 | 0.932 |
| With PPE[†] | | **0.865** | **0.788** | 0.934 | **0.983** | **0.676** | **0.534** | **0.783** | 0.934 |
| DistMult-test | 32 | 0.812 | 0.722 | 0.883 | 0.970 | 0.704 | 0.569 | 0.796 | 0.958 |
| With PPE | | **0.839** | **0.760** | 0.896 | **0.974** | 0.713 | 0.584 | 0.801 | 0.957 |
| With PPE[†] | | 0.837 | 0.759 | **0.898** | **0.974** | **0.714** | **0.585** | **0.803** | **0.959** |
| DistMult-test | 64 | 0.673 | 0.539 | 0.760 | 0.940 | 0.602 | 0.437 | 0.717 | 0.924 |
| With PPE | | 0.680 | 0.551 | 0.759 | 0.939 | **0.632** | **0.483** | 0.724 | **0.932** |
| With PPE[†] | | **0.686** | **0.554** | **0.777** | **0.946** | 0.629 | 0.475 | **0.724** | **0.932** |
| DistMult-test | 128 | 0.665 | 0.527 | **0.767** | **0.934** | 0.481 | 0.307 | 0.561 | 0.884 |
| With PPE | | 0.667 | **0.532** | 0.766 | 0.933 | 0.507 | 0.338 | 0.591 | 0.889 |
| With PPE[†] | | **0.669** | 0.531 | 0.766 | 0.933 | **0.520** | **0.351** | **0.600** | **0.890** |
| DistMult-test | 256 | 0.648 | 0.503 | 0.746 | 0.932 | 0.444 | 0.270 | 0.518 | 0.851 |
| With PPE | | 0.651 | 0.506 | 0.746 | 0.933 | 0.464 | 0.291 | 0.539 | 0.861 |
| With PPE[†] | | **0.659** | **0.511** | **0.765** | **0.936** | **0.475** | **0.309** | **0.541** | **0.868** |

of parameters is a promising means to improve the link prediction performance across models and datasets. On each epoch interval, our approach (PPE) updates the parameter ensemble model via ensemble weights with a running knowledge graph embedding model. Our experiments show that PPE constructs a high performing parameter ensemble *with virtually an expense of training a single model*.

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

# A   APPENDIX

# B   DETERMINING ENSEMBLE SEIGHTS

Figure 3 visualizes the ensemble weights with different growth rates. Using a growth rate of 1.0 implies that $\mathbf{w}_{\text{PPE}}$ is constructed with positive equal weights.

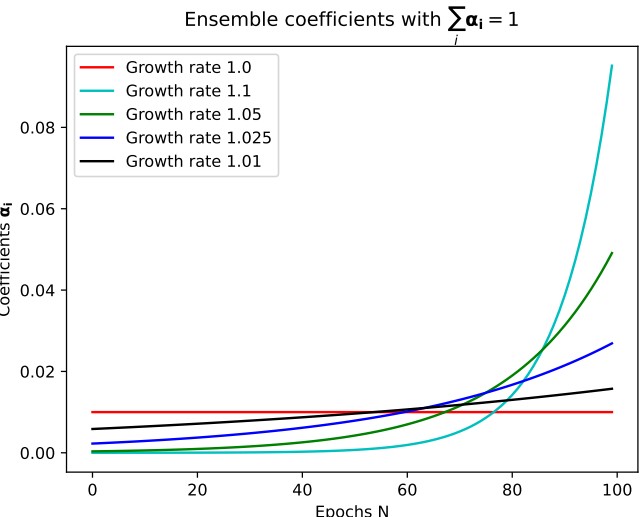

Figure 3: Ensemble weights with different growth rates.

## B.1   IMPLEMENTATION DETAILS AND REPRODUCIBILITY

We open-source our code including training and evaluation scripts at an anonymous project page.[1] Throughout our experiments, we used the same seed for the random number generator. All experiments are conducted on the same hardware.

## B.2   ADDITIONAL EXPERIMENTS

## B.3   DATASETS

KINSHIP describes the 26 different kinship relations of the Alyawarra tribe and the unified medical language system (UMLS) describes 135 medical entities via 49 relations describing (Trouillon

---

[1] https://drive.google.com/drive/folders/1jQo6FJgObyVaEMmIxj7QrVMkX5Jd9NDk?usp=share_link

Table 9: Link prediction results on the train, validation and test splits of UMLS and KINSHIP benchmark datasets. Bold results indicate the best results.

| | UMLS | | | | KINSHIP | | | |
|---|---|---|---|---|---|---|---|---|
| | MRR | @1 | @3 | @10 | MRR | @1 | @3 | @10 |
| DistMult-train | 0.992 | 0.987 | 0.997 | **1.000** | 0.847 | 0.781 | 0.893 | 0.977 |
| With PPE | **0.999** | **0.998** | **0.999** | **1.000** | **0.865** | **0.806** | **0.906** | **0.981** |
| DistMult-val | 0.458 | 0.325 | 0.500 | 0.753 | 0.399 | 0.256 | 0.432 | 0.741 |
| With PPE | **0.499** | **0.376** | **0.528** | **0.778** | **0.426** | **0.288** | **0.455** | **0.760** |
| DistMult-test | 0.450 | 0.321 | 0.491 | 0.755 | 0.404 | 0.260 | 0.442 | 0.755 |
| With PPE | **0.493** | **0.372** | **0.526** | **0.778** | **0.433** | **0.290** | **0.470** | **0.782** |
| ComplEx-train | 0.998 | 0.997 | **1.000** | **1.000** | 0.993 | 0.989 | 0.998 | **1.000** |
| With PPE | **1.000** | **1.000** | **1.000** | **1.000** | **0.996** | **0.993** | **0.999** | **1.000** |
| ComplEx-val | 0.442 | 0.285 | 0.521 | 0.766 | 0.521 | 0.378 | 0.595 | 0.829 |
| With PPE | **0.491** | **0.350** | **0.550** | **0.783** | **0.599** | **0.463** | **0.677** | **0.861** |
| ComplEx-test | 0.444 | 0.287 | 0.528 | 0.773 | 0.533 | 0.388 | 0.614 | 0.821 |
| With PPE | **0.502** | **0.361** | **0.573** | **0.787** | **0.603** | **0.479** | **0.675** | **0.842** |
| QMult-train | 0.998 | 0.998 | 0.999 | 0.999 | 0.990 | 0.983 | 0.996 | **0.999** |
| With PPE | **1.000** | **1.000** | **1.000** | **1.000** | **0.995** | **0.992** | **0.998** | **0.999** |
| QMult-val | 0.445 | 0.280 | 0.524 | 0.791 | 0.500 | 0.352 | 0.571 | 0.805 |
| With PPE | **0.485** | **0.326** | **0.578** | **0.803** | **0.598** | **0.468** | **0.675** | **0.852** |
| QMult-test | 0.426 | 0.272 | 0.498 | 0.757 | 0.502 | 0.355 | 0.580 | 0.801 |
| With PPE | **0.480** | **0.334** | **0.555** | **0.786** | **0.591** | **0.467** | **0.668** | **0.838** |

et al., 2017). FB15K-237 and YAGO3-10 are subsets of Freebase and YAGO (Dettmers et al., 2018), Never-Ending Language Learning datasets are designed to multi-hop reasoning capabilities released (NELL-995 h25, NELL-995 h50, NELL-995 h100) (Xiong et al., 2017). We were also interested to evaluate PPE on other benchmark datasets. So, we include Mutagenesis and Carcinogenesis benchmark datasets that are often used to benchmark concept learning in description logics (Heindorf et al., 2022).

