# OpenReview forum: "Polyak Parameter Ensemble: Exponential Parameter Growth Leads to Better Generalization"
_ICLR.cc/2024/Conference — Submitted to ICLR 2024_

### Official Review · Reviewer_cGaK · 2023-10-12

**Soundness:** 2 fair
**Presentation:** 2 fair
**Contribution:** 1 poor
**Rating:** 3
**Confidence:** 4

**Summary:**

The paper presents PPE (Polyak Parameter Ensemble), a technique more commonly known as Exponential Moving Average or Stochastic Weight Average. This tackles common ensembling flaws such as training overhead, inference overhead, and high memory needs. PPE collects checkpoints along training to build an interpolated weight average, enhancing performance. They experiment with varying interpolation coefficients and evaluate the method on the CIFAR dataset as well as in the context of link prediction and multi-hop query answering tasks.

**Strengths:**

- The work addresses weight averaging, an important technique to improve the reliability and robustness in deep learning. Works that empirically assess its benefits or theoretically understands it are valuable.
- They use multiple datasets for evaluation and the gains are consistent.

**Weaknesses:**

- The paper doesn't introduce a new strategy. The proposed PPE is very similar to all existing weight averaging strategies, previously described in several referenced works, listed below.

[Cha2021] SWAD: Domain Generalization by Seeking Flat Minima
[Arpit2022] Ensemble of Averages: Improving Model Selection and Boosting Performance in Domain Generalization
[Kaddour2022] Stop Wasting My Time! Saving Days of ImageNet and BERT Training with Latest Weight Averaging
[Sanyal2022] Understanding the Effectiveness of Early Weight Averaging for Training Large Language Models
[Li2023] Trainable Weight Averaging: Efficient Training by Optimizing Historical Solutions
[Wortsman2022a] Robust fine-tuning of zero-shot models
[Wortsman2022b] Model soups: averaging weights of multiple fine-tuned models improves accuracy without increasing inference time
[Ruppert1989] Efficient estimations from a slowly convergent robbins-monro process.

- The related work section appears deficient as it does not list the works above. While there's considerable discussion on dropout, several essential weight averaging strategies have been omitted.

- The experimental section doesn't sufficiently compare PPE with the other weight averaging or efficient ensembling techniques (such as dropout).

- The paper's contributions are not clearly stated and are not obvious. The sole contribution I can identify is applying weight averaging to new tasks like link prediction and multi-hop query answering. More generally, the introduction does not introduce sufficiently the paper.

**Questions:**

- What are the contributions of the paper, especially given that weight averaging is not new?
- From your experiments, how does the choice of the interpolating coefficient impact the final performances?
- Do you have any idea/intuition on why the benefits dissipate for very low embedding vector?
- Could you discuss the similarity and difference with the concurrent work "[Busbridge2023] How to Scale Your EMA"?

---

### Official Review · Reviewer_kkKM · 2023-10-20

**Soundness:** 2 fair
**Presentation:** 2 fair
**Contribution:** 1 poor
**Rating:** 3
**Confidence:** 4

**Summary:**

This work proposes a method called Polyak Parameter Ensembles which averages the parameters of a neural network over epochs. The authors suggest several methods for aggregating weights including taking an exponential average or a simple mean. The method is compared to standard training on the tasks of link prediction and multi-hop reasoning.

**Strengths:**

* The idea of averaging a models parameters across epochs is simple and effective.
* The experimental results cover an extensive number of datasets with granular evaluations performed within each.
* I have not seen this method applied to these tasks before which may provide useful empirical evidence.

**Weaknesses:**

* As far as I can tell, the proposed method is identical (or almost identical) to stochastic weight averaging [1,2] and descendants of this method. This seems to have been missed in the literature review (although, bizarrely, [1] was cited in a different context in the paper). Therefore this paper appears to be a clear reject.
* In Fig 2 and on P 3 the author’s attempt to argue that parameters will “hover/circle” around the global optimum. This is a subtly different claim to what previous works have claimed. This is only certainly true in 1d (I.e. fig 2). In high dimensional problems it is likely possible to reach arbitrarily close to the optimum in infinite time.
* There are several presentation issues:
   - eqn (1) switches between $\theta$ and w notation.
   - eqn (6) has errors in indexing which should start at 1.
   - $\alpha$ should not be bold if it is a scalar. Furthermore, $\odot$ typically denotes the Hadamard product which is not appropriate here.
   - Several typos such as “stooping” in Sec 3.1 and “seights” in the title of App B.
* Related work is lacking a broad context on modern deep ensemble and parameter averaging methods.
* Results on CIFAR-10 are promised but as far as I can tell they are only included in Fig 1.


[1] Averaging Weights Leads to Wider Optima and Better Generalization; Pavel Izmailov, Dmitry Podoprikhin, Timur Garipov, Dmitry Vetrov, Andrew Gordon Wilson; Uncertainty in Artificial Intelligence (UAI), 2018

[2] There Are Many Consistent Explanations of Unlabeled Data: Why You Should Average; Ben Athiwaratkun, Marc Finzi, Pavel Izmailov, Andrew Gordon Wilson; International Conference on Learning Representations (ICLR), 2019

**Questions:**

For any reasonable definition of an ensemble, averaging parameters throughout training doesn’t seem like a good fit. Could the authors explain in what meaningful way they view this as ensembling?

---

### Official Review · Reviewer_XnC1 · 2023-10-24

**Soundness:** 1 poor
**Presentation:** 2 fair
**Contribution:** 1 poor
**Rating:** 3
**Confidence:** 3

**Summary:**

The manuscript proposes an approach named PPE, by maintaining a running weighted average of the model parameters at each epoch interval, to alleviate the issues of computational overhead, increased latency, and memory requirements while improving the generalization performance.

The manuscript specifically considers knowledge graph embedding models and only evaluates the proposed methods for the link prediction task.

**Strengths:**

* The manuscript studies a crucial problem in the field.
* The method introduced in this manuscript is easy to understand.
* Several benchmark datasets are included for the evaluation.

**Weaknesses:**

1. Limited novelty with insufficient supports
    * Though this manuscript studies a crucial problem in the field, the proposed method is not novel.
    * The related work does not include the recent line of research on model averaging, e.g., [1, 2, 3, 4, 5], and fails to compare with them.
    * Only the link prediction task is evaluated.
    * Some simple baselines, e.g., tail-averaging, t-averaging, exponential averaging, should be included.
2. No theoretical supports/justifications can be found.
3. Minor comments:
    * The significance of considering the knowledge graph embedding model and the link prediction task should be explained. E.g., can we identify some unique challenges that cannot be addressed by existing solutions?
    * Table 1 is hard to understand, and it is unclear why we need to include it.


### Reference
[1] Stop wasting my time! saving days of imagenet and bert training with latest weight averaging, https://arxiv.org/abs/2209.14981

[2] Rethinking the inception architecture for computer vision, https://arxiv.org/abs/1512.00567

[3] Model soups: averaging weights of multiple fine-tuned models improves accuracy without increasing inference time, https://arxiv.org/abs/2203.05482

[4] How to Scale Your EMA, http://arxiv.org/abs/2307.13813

[5] Trainable Weight Averaging: Efficient Training by Optimizing Historical Solutions, https://openreview.net/forum?id=8wbnpOJY-f

**Questions:**

NA

---

### Official Review · Reviewer_2Kw3 · 2023-10-30

**Soundness:** 3 good
**Presentation:** 3 good
**Contribution:** 1 poor
**Rating:** 3
**Confidence:** 4

**Summary:**

This paper proposes Polyak Parameter Ensemble, an efficient technique for building parameter space ensembling.  It maintains an exponentially weighted average of parameters over training epochs to construct an ensemble. The weight for each ensemble component can be automatically determined using a validation set. Experiments demonstrate improvements in generalization in several knowledge graph tasks.

**Strengths:**

The method for determining the ensembling weight is a novel and interesting contribution.

The experiment details are provided, allowing for better reproducibility.

**Weaknesses:**

- The idea of averaging over weights during training has been proposed in [1], and is already implemented in PyTorch. Although the author does discuss [1], but the discussion should be way more detailed instead of just having: "Ensemble learning have been extensively studied in the literature (Bishop & Nasrabadi, 2006; Murphy, 2012; Huang et al., 2017; **Izmailov et al., 2018**)."

- The experiment sections lack baselines. I agree that polyak averaging style ensembling is more efficient than ensembling over multiple models. But how does it compare with methods, such as Deep Ensemble, in terms of performance?

- I can find Figure. 2 online: https://coffee-g9.hatenablog.com/entry/Polyak_averaging , I believe a proper citation should be added.


[1] Averaging Weights Leads to Wider Optima and Better Generalization; Pavel Izmailov, Dmitry Podoprikhin, Timur Garipov, Dmitry Vetrov, Andrew Gordon Wilson; Uncertainty in Artificial Intelligence (UAI), 2018

**Questions:**

Why is the CIFAR-10 performance in Figure. 1 so poor? The test accuracy is ~60, which is far lower than the number from modern networks such as ResNet 18, which can easily achieve a test accuracy of > 80.

---

### Meta-Review · Area_Chair_AW63 · 2023-11-27

**Metareview:**

This paper about the construction of parameter-space ensembles was shown to suffer from many weaknesses, for example:

* Lack of comparisons with important baselines
* Insufficient related work, in particular, it is unclear how the proposed method differs from SWA
* No theoretical grounding of the approach
* Clarity of the manuscript
* Unclear set of contributions

Since the author(s) did not prepare any rebuttal to try to address those concerns, and given the alignment of the reviewers' scores, the paper is recommended for a rejection.

**Justification For Why Not Higher Score:**

* Lack of comparisons with important baselines
* Insufficient related work
* Unclear differences with respect to SWA
* No theoretical grounding of the approach
* Clarity of the manuscript
* Unclear set of contributions

(and no rebuttal was provided to reply to the concerns above)

**Justification For Why Not Lower Score:**

N/A

---

### Decision · Program_Chairs · 2024-01-16

Reject